# Identification of a Novel Bioactive Peptide Derived from Frozen Chicken Breast Hydrolysate and the Utilization of Hydrolysates as Biopreservatives

**DOI:** 10.3390/biology12091218

**Published:** 2023-09-08

**Authors:** Mohamed Abdelfattah Maky, Takeshi Zendo

**Affiliations:** 1Laboratory of Microbial Technology, Division of Systems Bioengineering, Department of Bioscience and Biotechnology, Faculty of Agriculture, Graduate School, Kyushu University, 744 Motooka, Nishi-ku, Fukuoka 819-0395, Japan; mohamedmekky@vet.svu.edu.eg; 2Department of Food Hygiene and Control, Faculty of Veterinary Medicine, South Valley University, Qena 83523, Egypt

**Keywords:** bioactive peptides, chicken breast, angiotensin-converting enzyme, antioxidant, antimicrobial peptides, biopreservatives

## Abstract

**Simple Summary:**

The development of food-derived bioactive peptides with beneficial health effects has gained a lot of attention and is regarded as a natural approach to enhancing human health. Bioactive peptides are short protein fragments with beneficial biological functions. Frozen chicken is a protein-rich food that contains bioactive peptides, which can be produced through enzymatic hydrolysis. The current study was designed to purify and characterize bioactive peptides from frozen chicken breast. The obtained findings showed that frozen chicken breast is a promising source of bioactive peptides with inhibitory action on the angiotensin-converting enzyme and antimicrobial and antioxidant properties. Furthermore, chicken hydrolysates could significantly reduce bacterial growth and lipid oxidation when applied to chicken breast. Muscle-derived hydrolysates, particularly chicken hydrolysate, hold promise as potential natural preservatives to replace chemical additives.

**Abstract:**

Frozen chicken breast was hydrolyzed by treatment with thermolysin enzyme to obtain a chicken hydrolysate containing bioactive peptides. After that, a peptide was purified from the chicken hydrolysate utilizing a Sep-Pak C18 cartridge and reversed-phase high-performance liquid chromatography (RP-HPLC). The molecular weight of the chicken peptide was 2766.8. Protein sequence analysis showed that the peptide was composed of 25 amino acid residues. The peptide, designated as C25, demonstrated an inhibitory action on the angiotensin-converting enzyme (ACE) with a half maximal inhibitory concentration (IC_50_) value of 1.11 µg/mL. Interestingly, C25 showed antimicrobial activity against multi-drug resistant bacteria *Proteus vulgaris* F24B and *Escherichia coli* JM109, both with MIC values of 24 µg/mL. The chicken hydrolysate showed antioxidant activity with an IC_50_ value of 348.67 µg/mL. Furthermore, the proliferation of aerobic bacteria and *Enterobacteriaceae* as well as lipid oxidation were significantly reduced when the chicken hydrolysate was used as a natural preservative during cold storage of chicken breasts. Hydrolysates derived from muscle sources have the potential to be used in formulated food products and to contribute positively to human health.

## 1. Introduction

Bioactive peptides are small protein fragments that have positive effects on human health. There are many ways to produce bioactive peptides, including food fermentation and enzymatic hydrolysis [1]. Bioactive peptides have received great attention around the world due to their medicinal advantages as well as their normal and healthy properties. Bioactive peptides are generated from parent proteins by proteolytic enzymes, resulting in peptides of different lengths and variable structural configurations, which may also be associated with activity [2]. Hence, different proteases can cleave proteins at various locations, resulting in a large number of peptide fragments with different degrees of bioactivity [3]. Hydrolysates can be purified by liquid chromatography, and the purified fractions showing biological activity can be structurally analyzed using a protein sequencer and a mass spectrometer. Structural data on the newly identified bioactive peptides are important and valuable to further investigate the relationship between structure and activity by comparing reported bioactive peptides on the database.

Food-derived peptides have been investigated for their beneficial impacts, including immunomodulatory, antibacterial, antioxidant, dipeptidyl peptidase-IV inhibitory, antiviral, anti-inflammatory and angiotensin-converting enzyme (ACE) inhibitory properties [4,5]. Chemical preservatives have been utilized in foods to prevent the growth of bacteria. However, consumers have experienced negative health impacts from the addition of chemicals. Food biopreservation is a promising technology to produce food with a low chemical content by utilization of naturally occurring antimicrobial substances in food. Food-derived bioactive peptides are prospective biopreservatives that can replace harmful chemical preservatives [6,7]. Peptides produced from food proteins can be safe natural alternatives to synthetic molecules [8].

Several studies have been conducted on the generation of bioactive peptides from animal proteins. Many of them are based on dairy proteins, which tend to have different physiological properties [9]. Bioactive peptides have also been extracted from meat [10,11], duck breast [12], and fish [13,14]. Among animal foods, chicken meat is highly demanded by consumers around the world. It is regarded as a valuable source of high-quality protein, rich in necessary amino acids and bioactive peptides [15]. Sangsawad et al. [16] reported three new bioactive peptides generated from cooked chicken meat after enzymatic hydrolysis which exhibited ACE inhibitory activities.

Studies have been carried out on the efficacy of protein hydrolysates generated from various sources as biopreservatives. Wang et al. [17] reported that the application of mutton hydrolysates to mutton burgers enhanced their oxidative stability. In addition, milk hydrolysates can inhibit the oxidation of lipids in muscle foods [18]. The addition of pig blood hydrolysates to pork emulsion slows down lipid oxidation and bacterial proliferation [19]. The advantages of using bioactive peptides as an additive to food have been addressed by Wolfe and Liu [20]. To date, there is a lack of studies investigating the efficacy of chicken breast hydrolysates as biopreservatives.

Previously, we have successfully generated bioactive peptides from fish (F21) and beef skeletal muscle (B34). They have been purified, identified and characterized. Both F21 and B34 exhibited antibacterial and ACE inhibitory action [14]. However, there is a paucity of data on the extraction, purification, identification and characterization of bioactive peptides from frozen chicken breasts. Hence, the first aim of the current work was to purify a bioactive peptide generated from frozen chicken breast and investigate its structural and biological characteristics. Secondly, the potential application of chicken, beef and fish hydrolysates as natural food preservatives was studied on the chicken breast during food storage.

## 2. Materials and Methods

### 2.1. Extraction of Protein by Enzymatic Hydrolysis

The frozen chicken breast was subjected to extraction of bioactive peptides according to the method reported by Jang and Lee [4] with a small modification. Briefly, 30 g of frozen chicken breast was mixed with 200 mL of 0.02 M sodium phosphate buffer (pH 7.4) and centrifuged at 15,770× *g* for 20 min at 4 °C. The supernatant was collected, and the pH was adjusted to 7.5 and mixed with 10 mg of thermolysin enzyme from *Bacillus thermoproteolyticus* Rokko (EC No. 232-973-4; CAS No. 9073-78-3; Sigma-Aldrich, St. Louis, MO, USA). Following 8 h of digestion at 37 °C, the preparation was boiled for 5 min to inactivate the enzyme and then it was cooled to room temperature.

### 2.2. Purification of Chicken Breast Hydrolysate

The thermolysin-digested chicken meat was concentrated using a Speed Vac concentrator (Savant, Farmingdale, NY, USA) and sterilized via a sterile cellulose acetate membrane filter (0.2 µm, Advantec, Tokyo, Japan) and was referred to as the chicken hydrolysate. Peptides in the hydrolysate were recovered using a Sep-Pak C_18_ cartridge (Waters, Milford, MA, USA) and eluted with acetonitrile containing 0.1% trifluoroacetic acid (TFA). To eliminate the solvent, the eluted fractions were run through a Speed Vac concentrator. The fraction was then transferred to an Atlantis dC18 column (4.6 mm × 150 mm, 5 µm; Waters) in an LC-2000 Plus high-performance liquid chromatography (HPLC) system (JASCO, Tokyo, Japan). The elution procedure used a gradient of Milli-Q-acetonitrile containing 0.1% TFA at a flow rate of 1 mL/min: 0–45 min, 0–70% acetonitrile. A Pierce bicinchoninic acid (BCA) protein assay kit was used to measure the protein concentration in the hydrolysates and the purified peptides (Thermo Fisher Scientific, Waltham, MA, USA).

### 2.3. Mass Spectrometry and Amino Acid Sequencing

The molecular weights of the chicken peptides were determined by electrospray ionization-time of flight mass spectrometry (ESI-TOF MS) using a JMST100LC mass spectrometer (JEOL, Tokyo, Japan) as described by Zendo et al. [21]. For detailed studies, the purest fraction from the chicken hydrolysate that appeared in the mass spectrum was chosen for further analyses. The molecular mass of the bioactive peptide was determined by electrospray ionization-time of flight mass spectrometry (ESI-TOF MS) using a JMS-T100LC mass spectrometer (JEOL, Tokyo, Japan). Data acquisition was performed for the obtained mass spectrum using the JEOL MassCenter program (JEOL).

Edman degradation with the PPSQ-31 protein sequencer (Shimadzu, Kyoto, Japan) was used to determine the amino acid sequence of the purified chicken peptide termed C25 based on the manufacturer’s directions [22]. Briefly, the purified chicken peptide was applied to the protein sequencer. The resulting data for each cycle of Edman degradation were analyzed utilizing the PPSQ-31 program and the amino acid sequences were identified.

### 2.4. Sequences Analysis

The amino acid sequence of C25 was analyzed utilizing the BLAST program of the National Center for Biotechnology Information (NCBI) (http://www.ncbi.nlm.gov/BLAST, accessed on 7 June 2023). The GOR technique accessible via Prabiserver (https://npsa-prabi.ibcp.fr, accessed on 7 June 2023) was used to estimate the secondary structure.

### 2.5. ACE Inhibition Activity Assay

The chicken hydrolysate and C25 were tested for their ability to inhibit ACE utilizing the ACE kit-WST (Dojindo Laboratories, Kumamoto, Japan) based on the manufacturer’s instructions and as described in detail by Maky and Zendo [14]. Briefly, 20 µL of the assayed sample was applied to the microtiter plate, followed by the addition of substrate buffer and other kit reagents as described in the instructions. The microtiter plate was incubated at 37 °C for one hour. Then, 200 µL of indicator solution was added, and the plate was incubated for 10 min at room temperature. Using a Sunrise microplate absorbance reader (Tecan), the absorbance at 450 nm was recorded.

### 2.6. Antioxidant Activity Assay

The antioxidant activity of the chicken hydrolysate and C25 was assessed utilizing a DPPH antioxidant assay kit (Dojindo Laboratories) as described in the instructions provided by the manufacturer and as elaborated by Maky and Zendo [14]. Briefly, 20 µL of the investigated samples was applied to a microtiter plate and 80 µL of assay buffer, 100 µL of ethanol and 100 µL of DPPH working solution were mixed with the samples. Then, the plate was incubated at 25 °C for 30 min in the dark. The absorbance was measured by using a Sunrise microplate absorbance reader (Tecan) at 450 nm.

### 2.7. Determination of Antimicrobial Activity

The following indicator bacteria were used to determine the minimum inhibitory concentrations (MICs) as described by Wiegand et al. [23] and Maky and Zendo [14] by using a broth microdilution assay. *Weizmannia* (*Bacillus*) *coagulans* JCM 2257^T^, *Salmonella enterica* serovar Typhimurium NBRC 13243^T^, *Listeria innocua* ATCC 33090^T^, *Proteus vulgaris* F24B [24] and *Escherichia coli* JM109 were cultivated in Tryptic Soy Broth (BD, Sparks, MD, USA) supplemented with 0.6% yeast extract. *Enterococcus faecalis* JCM 5803^T^ was cultivated in MRS medium (Oxoid, Basingstoke, UK), while *Pseudomonas putida* ATCC 12633^T^ was cultivated in Luria Bertani medium (BD). Briefly, the bacterial inhibition was assessed at an optical density at 620 nm using an Infinite F200 Pro microplate reader (Tecan, Männedorf, Switzerland) after the indicator strains were cultured with twofold serial dilutions of chicken hydrolysates and C25.

### 2.8. Preservative Effects of Beef, Fish and Chicken Hydrolysates

To clarify the impacts of beef, fish and chicken hydrolysates on the bacteriological and chemical quality of chicken breast, the hydrolysates were applied directly to the surface of chicken breast. Four groups were prepared, each of 20 g. Briefly, the first group served as the untreated group, and the second, third and fourth groups were treated with 200 μg/g of the beef hydrolysate, the fish hydrolysate and the chicken hydrolysate, respectively. Both the beef and fish hydrolysates were prepared as previously described by Maky and Zendo [14]. Briefly, fish (*Gadidae*) and beef skeletal muscles were hydrolyzed with pepsin enzyme for 8 h. The digested fish and beef materials were sterilized by a sterile cellulose acetate membrane filter and designated as fish and beef hydrolysates, respectively. All groups were stored at 4 °C and samples (days 0, 4, 8) were collected for further analysis.

### 2.9. Evaluation of Antibacterial Activity of Hydrolysates on Bacterial Flora of Chicken Breast

Enumeration of aerobic bacteria, *Enterobacteriaceae* and lactic acid bacteria (LAB) of the samples was performed at different time intervals by using plate count agar, MacConkey agar and MRS, respectively [25]. Briefly, on days 0, 4 and 8, the samples were homogenized for 2 min in 0.1% peptone water (BD, Sparks, MD) at a ratio of 1:10, and decimal dilutions from 10^−1^ to 10^−5^ were made using the same diluents. An amount of 100 μL of each dilution was plated onto appropriate culture media, and the plates were incubated at 35 °C to enumerate colonies formed after 48 h [25].

### 2.10. Lipid Oxidation

The effects of the hydrolysates as antioxidant compounds on lipids were assessed utilizing thiobarbituric acid reactive substance (TBARS) as described by Khan et al. [26]. Briefly, 20 mL of trichloroacetic acid and 20 mL of distilled water were mixed with 5 g of the sample under investigation. The mixture was centrifuged at 4000× *g* for 5 min and the supernatant was collected and filtered. One milliliter of 0.01 M 2-thiobarbituric acid was mixed with four milliliters of the filtrate and boiled in a water bath for 25 min, and the absorbance of the resulting colored solution was measured at 532 nm using a Sunrise microplate absorbance reader (Tecan). TBARS values (represented as mg malonaldehyde equivalents/kg) were calculated by multiplying the absorbance reading by a factor of 10.2.

### 2.11. Statistical Analysis

Each experiment was performed in triplicate and the SPSS program was utilized for statistical analyses. Analysis was performed utilizing a one-way ANOVA, a comparison of averages was performed using Duncan’s multiple range tests and the significance was established at a level of *p* ≤ 0.05.

## 3. Results

### 3.1. Generation, Purification and Structural Analysis of the Chicken Peptide

The chicken breast was digested with thermolysin and the resulting hydrolysate was applied to a Sep-Pak C_18_ cartridge. The peptides were eluted with 30% acetonitrile and then purified by reverse-phase HPLC (Figure 1).

All RP-HPLC fractions were examined by mass spectrometry, and the fraction without impurities was chosen for detailed investigations. The peptide in this fraction was termed C25, and further purified by two additional HPLC runs. As a result, approximately 2 mg of C25 was obtained from 30 g of frozen chicken breast, and its molecular weight was determined by mass spectrometry to be 2766.8 (Figure 2).

Additionally, Edman degradation revealed the amino acid sequence of C25 to be IHAEEILDIRGNPTVEVDLHTAKGH. The observed molecular weight was almost identical to that calculated from the identified amino acid sequence. A database exploration revealed that C25 had a high similarity to partial sequences of beta-enolase enzymes [27]. The secondary structure was suggested by using the GOR method, which indicated that C25 potentially forms extended strands and random coils (Figure 3).

### 3.2. ACE Inhibition and Antioxidant Activities

C25 showed significant ACE inhibitory activity (IC_50_, 1.11 µg/mL), whereas the chicken hydrolysate had lower activity (IC_50_, 592.4 µg/mL). Antioxidant activity was detected in the chicken hydrolysate (IC_50_, 348.67 µg/mL; TEAC, 0.190), whereas no activity was observed in C25.

### 3.3. Antimicrobial Spectra

The antibacterial activities of the chicken hydrolysate and C25 were observed against certain indicator bacteria (Table 1). Interestingly, the chicken hydrolysate and C25 also showed antibacterial activity toward the multi-drug resistant bacterium *Proteus vulgaris* F24B.

### 3.4. Preservative Effect of Beef, Fish and Chicken Hydrolysates

#### 3.4.1. Microbial Analysis

Several bacterial counts were carried out to clarify how the addition of the hydrolysates affected the bacteriological state of the chicken breast. The chicken hydrolysate exhibited a significant decline (*p* ≤ 0.05) in the aerobic plate count compared to the control group (Figure 4).

The chicken hydrolysate was the most potent in reducing the *Enterobacteriaceae* count followed by the fish hydrolysate (Figure 5).

Furthermore, the fish hydrolysate had a significant effect (*p* ≤ 0.05) on the LAB count in the stored chicken breast sample (Figure 6).

#### 3.4.2. Lipid Oxidation

The impact of the addition of beef, fish and chicken hydrolysates to chicken breast on lipid oxidation was evaluated by conducting a TBARS assay. Lipid oxidation was efficiently suppressed at TBARS values in the sample treated with the chicken hydrolysate (Figure 7).

## 4. Discussion

The degradation of muscle food by proteolytic enzymes can be a promising approach for the generation of bioactive peptides with beneficial biological characteristics. Frozen muscle food can be a promising source of bioactive peptides, as freezing denatures proteins through several chemical and physical mechanisms [28], which boosts the production of bioactive peptides [29].

The current work is the first report of a novel peptide being produced from frozen chicken breast. Here, we reported the extraction, purification and identification of a novel bioactive peptide utilizing chromatographic separation, mass spectrometry and protein sequence analysis (Figure 1, Figure 2 and Figure 3). The generation and accessibility of bioactive peptides from muscle food can be impacted by freezing. Proteins are denatured by freezing as a result of many physical and chemical stress mechanisms, such as ice development, pH shifts and low temperature, resulting in an increased productivity of bioactive peptides [29].

The thermolysin enzyme was employed to digest chicken breasts for 8 h. Thermolysin is a heat-stable enzyme that breaks down hydrophobic and/or aromatic amino acids. (ACE) seems to favor substrates with hydrophobic and/or aromatic amino acids at the C-terminus. Hence, thermolysin is selected as a common enzyme to generate bioactive peptides from food [30]. Therefore, thermolysin has been used to generate various bioactive peptides from muscle protein [31].

The identified peptide, termed C25, was composed of 25 amino acid residues, and the observed molecular weight was close to the calculated molecular weight. These findings revealed that C25 lacked any post-translational modifications commonly found in antimicrobial peptides. The amino acid sequence of C25 was highly similar to a portion of beta-enolase, a glycolytic enzyme that promotes the reversible conversion of 2-phosphoglycerate to phosphoenolpyruvate [32]. Beta-enolase-related peptides have also been generated from Panxian, a traditional Chinese dry-cured ham [33]. To the best of our knowledge, the current study is the first work on the extraction of beta-enolase-related peptides from chicken breasts.

C25 demonstrated a significant ACE inhibitory activity. C25 showed a more potent ACE inhibitory activity (IC_50_ = 0.402 μM) than the peptides generated by pepsin digestion of boneless chicken leg meat, whose IC_50_ ranged from 5.5 to 228 μM [34]. In the same context, three ACE inhibitory peptides with shorter amino acid sequences than C25 (KPLLCS, ELFTT and KPLL) generated after in vitro gastrointestinal digestion of chicken meat using various techniques were less effective than C25, with IC_50_ values of 0.37, 6.35 and 11.98 μM, respectively [16]. The ACE inhibitory activity of C25 was more potent than VPP and IPP, marketed as dietary additives produced from Calpis (Japan), which have IC_50_ values of 9 and 5 μM, respectively [35]. In addition, C25 displayed a higher ACE inhibitory activity than F21 and B34, bioactive peptides generated after pepsin digestion of fish and beef skeletal muscle, respectively (IC_50_ values of 7.3 µg/mL and 5.8 µg/mL for F21 and B34, respectively), as characterized in our previous study [14]. Hence, C25 can be considered a promising compound for the development of dietary supplements for people with hypertension. Hydrophobic amino acids in the peptides can enhance the ACE inhibitory activity by improving the interaction of the peptides with the target enzyme as reported by Matsui and Tanaka [36]. Furthermore, the ACE inhibitory effect is influenced by the presence of positively charged amino acid residues in bioactive peptides including His, Lys and Arg [37]. The ACE inhibitory action of peptides is also affected by their molecular weight; a report showed that peptides with a molecular weight of less than 3 kDa demonstrated greater activity than larger peptides [37]. The structural properties of C25 satisfy these requirements and a high ACE inhibitory activity has been experimentally demonstrated.

Free radicals can be produced as a result of lipid oxidation, causing major health issues and having a negative impact on food quality. One of the solutions to prevent lipid oxidation is the use of protein hydrolysates. In the current work, the chicken hydrolysate displayed a considerable antioxidant activity. A lower activity was obtained by another chicken hydrolysate generated by papain digestion of chicken breast, which had an EC_50_ value of 1.28 mg/mL [15]. Furthermore, the chicken hydrolysate generated by thermolysin in the current study showed a greater antioxidant activity than fish and beef hydrolysates reported by Maky and Zendo [14], which displayed IC_50_ values of 470.4 µg/mL and 531.0 µg/mL, respectively. However, C25 showed no antioxidant activity in our study. Similarly, a beef hydrolysate peptide, B34, exhibited no antioxidant action activity [14]. These results demonstrated that the antioxidant activity of peptides is related to many factors, including structural properties, amino acid sequences, hydrophobicities and molecular weights [38,39]. Peptides with a molecular weight of 500–1500 Da have greater antioxidant activities than peptides with a molecular weight of more than 1500 Da [40]. This is most likely due to the fact that lysine and arginine residues have strong antioxidant properties [41]. In addition, the antioxidant capacity is influenced by the kind of peptidase used for hydrolysis [42]. In the same context, Xiong et al. [43] concluded that hydrolysates have a higher antioxidant activity than pure peptides, as chicken hydrolysate, a mixture of peptides, showed a higher antioxidant activity than a purified peptide, C25, in this study. Butylated hydroxyanisole (BHA) and butylated hydroxytoluene (BHT) are two common antioxidant compounds used commercially in foods. Their IC50 values of 112.05 µg/mL and 202.35 µg/mL, respectively [44], are slightly greater than those of chicken hydrolysate.

Food-derived peptides are a promising strategy to control harmful bacteria, especially those resistant to antibiotics [45]. Remarkably, antibacterial activity was demonstrated by the chicken hydrolysate and C25 against an antibiotic-resistant bacterium, *Proteus vulgaris* F24B (Table 1). B34 showed antibacterial activity against *Proteus vulgaris* F24B, while F21 displayed no activity [14]. Hydrolysates generated by the enzymatic hydrolysis of chicken liver only showed weak antimicrobial activity against *Micrococcus luteus* [46]. The antimicrobial activity of peptides is influenced by their secondary structure, charge and number of amino acid residues [47]. Less recognized studies have shown that meat-derived peptides have considerable antibacterial action [48]. Synthetic compounds used as food preservatives have broader antibacterial spectra compared to bioactive peptides. However, adverse health effects have been reported. The promising biological activity shown by beef, fish [14] and chicken hydrolysates encouraged us to utilize them as biopreservatives, to use them as natural food preservatives in the future. The impact of beef, fish and chicken hydrolysates as biopreservatives in chicken breasts was evaluated.

The total aerobic bacterial count (Figure 4) decreased significantly with the addition of chicken hydrolysate on day 4. The untreated group showed a greater microbial population. For the *Enterobacteriaceae* count (Figure 5), the most effective additive was chicken hydrolysate, which showed a significant change on day 4 but not on day 8. LAB counts were not significantly inhibited on day 4, but fish hydrolysate exhibited a considerable reduction on day 8. 

Lipid oxidation is a significant element associated with meat rancidity, and was determined by the TBARS assay. TBARS are generated in the second step of autoxidation, where peroxides are converted to malondialdehyde. All samples had increased TBARS values over the days of the experiment, except those treated with chicken hydrolysate, which was significantly reduced on day 4 (Figure 7). These findings demonstrate that chicken hydrolysate has a beneficial effect on the keeping quality of chicken breast by reducing lipid oxidation. Consequently, the chicken hydrolysate could be a possible contender to combat chicken rancidity. Notably, the addition of the hydrolysates did not affect the sensory qualities of the chicken breast.

To sum up, chicken hydrolysates showed interesting effects against microbial proliferation in chicken breasts stored at 4 °C. Our findings were consistent with previous reports. Przybylski et al. [49] reported that peptides extracted from slaughterhouse by-products were successful in reducing spoilage bacteria and lipid oxidation during meat storage. Catiau et al. [50] reported that a bioactive peptide generated from bovine cruor has the potential to be used as a preservative for meat and its products. Based on the best of our knowledge, this study is the first report on the application of beef, fish and chicken hydrolysates as a natural preservative material in chicken breasts.

## 5. Conclusions

In this study, chicken hydrolysate was obtained from frozen chicken breasts after digestion with thermolysin, from which a novel bioactive peptide C25 was extracted and identified. C25 displayed ACE inhibitory action and antimicrobial activity. Interestingly, the use of chicken hydrolysates as a natural preservative during the cold storage of chicken breast significantly inhibited the proliferation of some food microbes, as well as lipid oxidation. Hydrolysates generated from muscle, particularly chicken hydrolysate, are promising natural preservatives to replace chemical additives. The obtained results deepen our comprehension of poultry products’ potential as functional components in the food industry and pharmaceutical sector. Therefore, to maximize their biological activity and their applications, it is essential to evaluate the potential mechanisms of action that may be involved.

## Figures and Tables

**Figure 1 biology-12-01218-f001:**
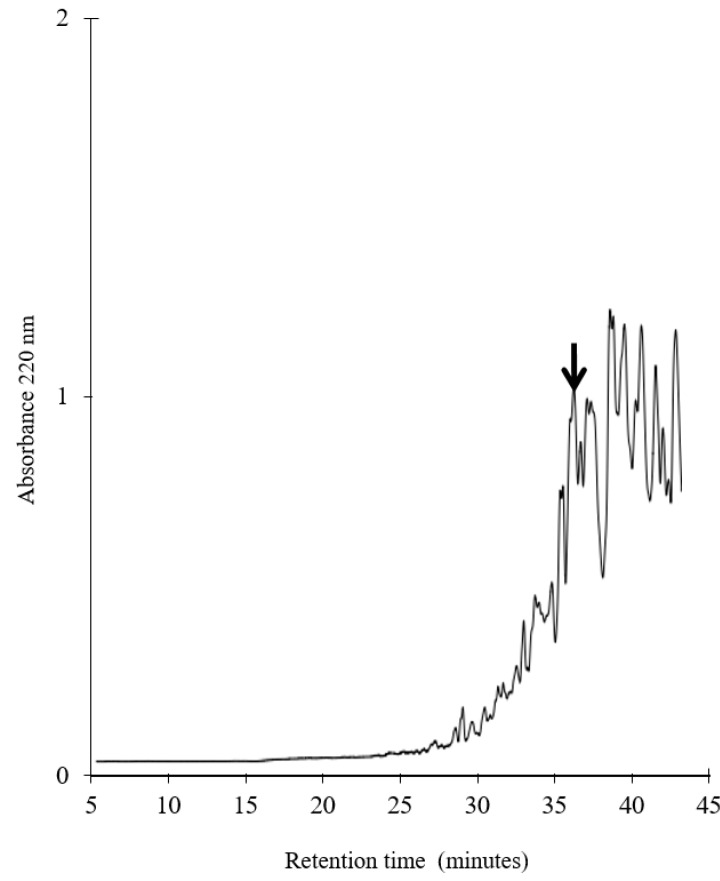
Purification of the bioactive peptide from frozen chicken breast. The chromatographic profile of the chicken hydrolysate was obtained by RP-HPLC with a dC18 column. A peak indicated by an arrow shows the target peptide.

**Figure 2 biology-12-01218-f002:**
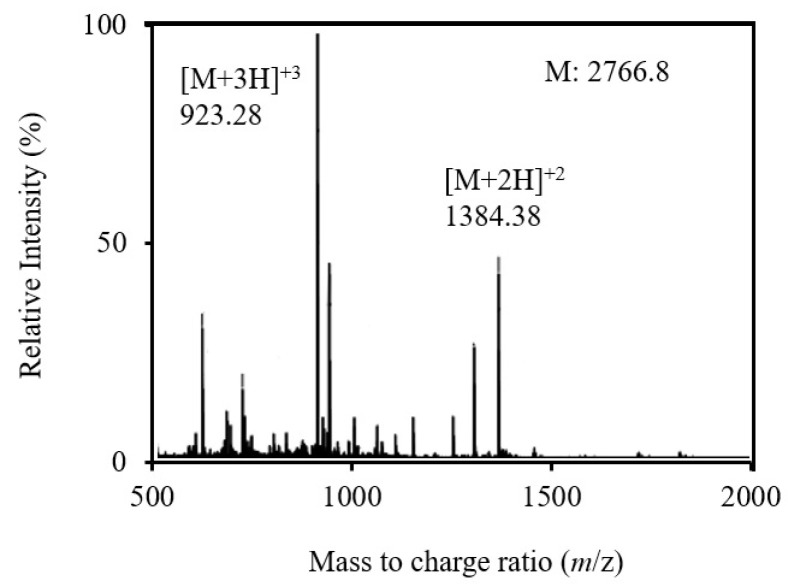
ESI-TOF mass spectrum of the purified chicken peptide. Multiple charged molecular ions of the peptide were detected and indicated. The molecular mass of the peptide was calculated based on the most abundant peak.

**Figure 3 biology-12-01218-f003:**
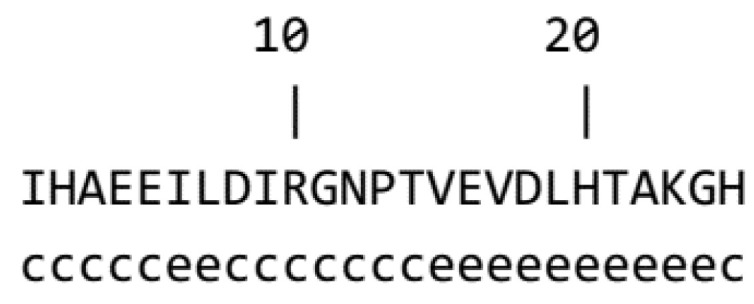
The prediction of the secondary structure of the chicken peptide. The prediction of the secondary structure was conducted by the GOR program on the Prabi server. c and e represent the secondary structure of random coil and exceeded strand, respectively.

**Figure 4 biology-12-01218-f004:**
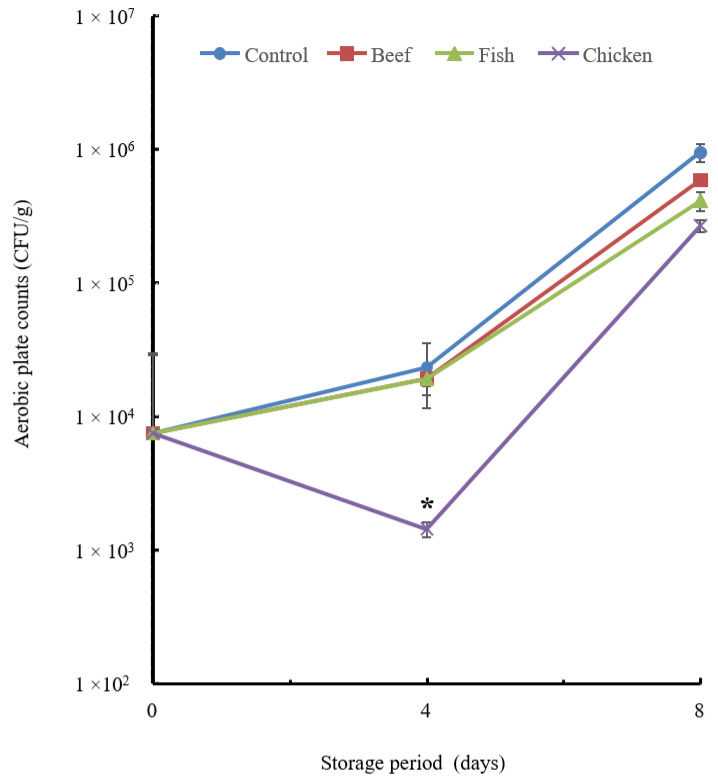
Aerobic plate count of the chicken breast treated with hydrolysate. Chicken breast samples were treated with beef, fish and chicken hydrolysates and then stored at 4 °C for 8 days. A sample without treatment served as a control. All the data are expressed as the mean + standard error derived from triplicate experiments. The chicken hydrolysate showed a significant difference at the *p* < 0.05 level in comparison with the control on day 4, indicated by asterisks.

**Figure 5 biology-12-01218-f005:**
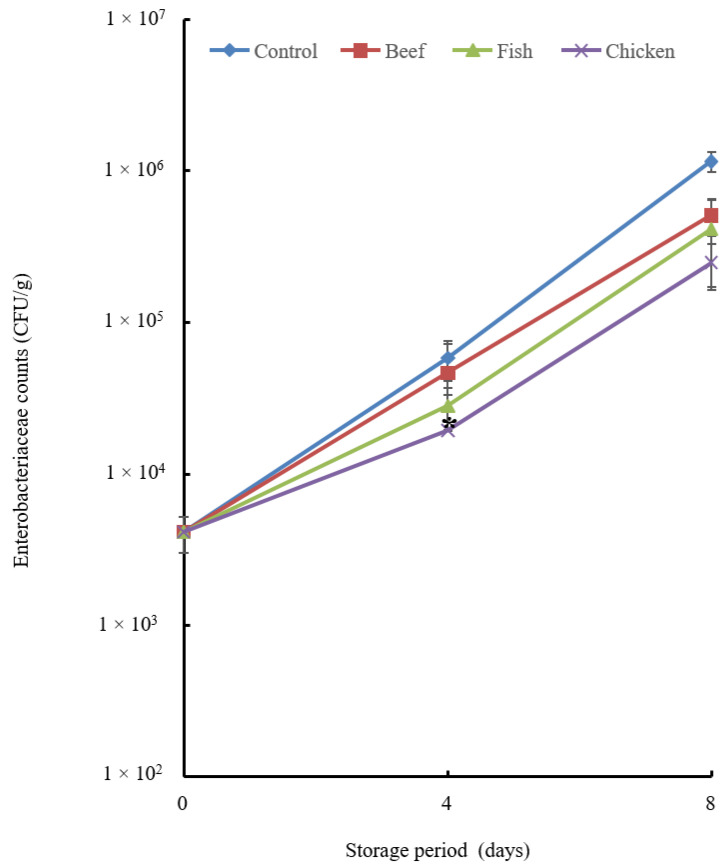
Enterobacteriaceae count of the chicken breast treated with hydrolysate. Chicken breast samples were treated with beef, fish and chicken hydrolysates and then stored at 4 °C for 8 days. A sample without treatment served as a control. Data are means + standard error derived from triplicate experiments. The chicken hydrolysate showed a significant difference at the *p* < 0.05 level in comparison with the control on day 4, indicated by asterisks.

**Figure 6 biology-12-01218-f006:**
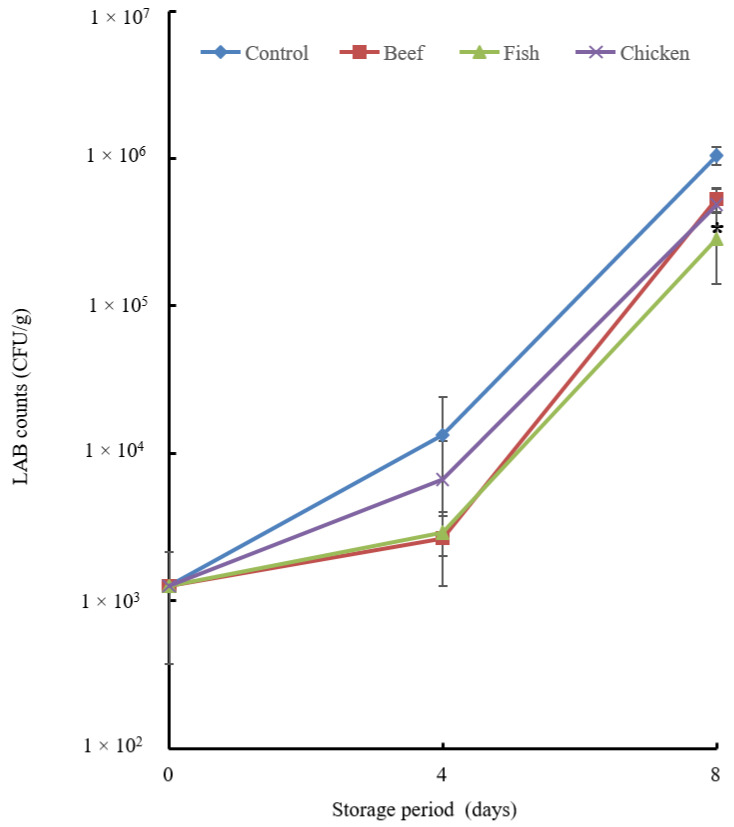
LAB count of the chicken breast treated with hydrolysate. Chicken breast samples were treated with beef, fish and chicken hydrolysates and then stored at 4 °C for 8 days. A sample without treatment served as a control. Data are means + standard error derived from triplicate experiments. The fish hydrolysate showed a significant difference at the *p* < 0.05 level in comparison with the control on day 8, indicated by asterisks.

**Figure 7 biology-12-01218-f007:**
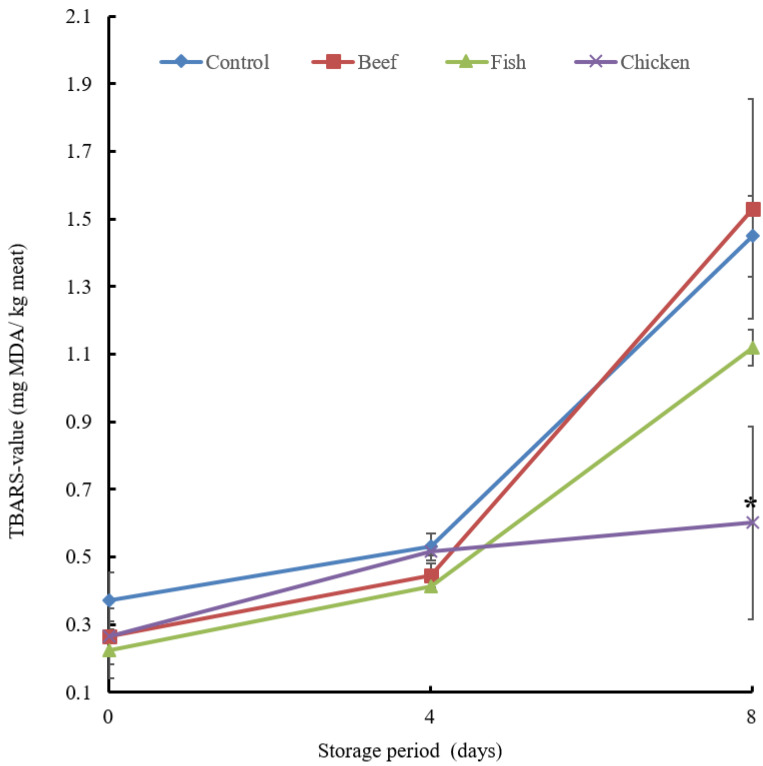
Changes in TBARS values in chicken breast treated with hydrolysate. Chicken breast samples were treated with beef, fish and chicken hydrolysates and then stored at 4 °C for 8 days. A sample without treatment served as a control. Data are means + standard error derived from triplicate experiments. The chicken showed a hydrolysate significant difference at the *p* < 0.05 level in comparison with the control on day 8, indicated by asterisks.

**Table 1 biology-12-01218-t001:** Antimicrobial spectra of the chicken hydrolysate and C25.

Indicator Strain	MIC (μg/mL)	
	Hydrolysate	C25
*Enterococcus faecalis* JCM 5803^T^	NA ^1^	NA ^2^
*Listeria innocua* ATCC 33090^T^	400	NA ^2^
*Escherichia coli* JM109	NA ^1^	24
*Weizmannia coagulans* JCM 2257^T^	NA ^1^	NA ^2^
*Pseudomonas putida* ATCC 12633^T^	800	NA ^2^
*Salmonella enterica* serovar Typhimurium NBRC 13245^T^	NA ^1^	NA ^2^
*Proteus vulgaris* F24-B	100	24

JCM, Japan Collection of Microorganisms (Wako, Japan); ATCC, American Type Culture Collection (Manassas, VA); NBRC, National Institute of Technology and Evaluation Biological Resource Center (Chiba, Japan). NA ^1^ (no activity) > 800 μg/mL, NA ^2^ > 24 μg/mL.

## Data Availability

The data presented in this study are available on request from the corresponding author.

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
