# Peer review of "Identification of a Novel Bioactive Peptide Derived from Frozen Chicken Breast Hydrolysate and the Utilization of Hydrolysates as Biopreservatives"

_biology, 2023, doi:10.3390/biology12091218_

Round 1
Reviewer 1 Report
Comments:
Title: Identification of a novel bioactive peptide derived from frozen chicken breast hydrolysate and the utilization of hydrolysates as biopreservatives
The manuscript has some interesting findings, and frozen chicken breast hydrolysate can be utilized as biopreservatives. However, the manuscript needs the clarifications from the following points, and also the writing should be polished to make it easier read for other researchers.
1. The title highlighted the utilization of frozen chicken breast hydrolysate as biopreservatives, but in the section of introduction, the author summarized the antioxidant, ACE inhibition, lipid oxidation, and antimicrobial properties. It should be clear and clarify the interlink between biopreservatives and these bioactivities in this part.
2. The introduction part summarized the normal production and bioactivities of bioactive peptides, and lack of the corresponding reviews. Protein hydrolysates/peptides have already been used in biopreservation? Peptides from which protein sources are widely used? The research status of chicken breast as biopreservatives.
3. I noticed that current research used thermolysin to hydrolyze chicken breast, why this enzyme, what the basis
4. In this study, why chosen beef and fish hydrolysates as the control to compare the preservative effects rather than some commercial preservatives? Did beef and fish hydrolysates have been reported as promising preservative agents.
5. The quality of pictures should be improved to enhance the readability of the manuscript
6. In lines 26, please change the “pure chicken peptide” as “chicken peptide”
Reviewer 2 Report
In the manuscript, a peptide was purified from the chicken hydrolysate utilizing a Sep-Pak C18 cartridge and reversed-phase high-performance liquid chromatography (RP-HPLC). The activities of C25 were explored. The manuscript can not be accept in its current form.
1. The introduction cannot highlight the latest research progress, nor can it reflect the innovation and significance of the manuscript.
2. The antioxidant activity was detected by only one method, in addition to which other methods were used for further detection.
3. What is the yield of C25?
4. The article should choose one activity to study in depth and explain its mechanism.
5. Molecular docking can be used to study the mechanism of activity.
6. The language of the article needs improvement.
In the manuscript, a peptide was purified from the chicken hydrolysate utilizing a Sep-Pak C18 cartridge and reversed-phase high-performance liquid chromatography (RP-HPLC). The activities of C25 were explored. The manuscript can not be accept in its current form.
1. The introduction cannot highlight the latest research progress, nor can it reflect the innovation and significance of the manuscript.
2. The antioxidant activity was detected by only one method, in addition to which other methods were used for further detection.
3. What is the yield of C25?
4. The article should choose one activity to study in depth and explain its mechanism.
5. Molecular docking can be used to study the mechanism of activity.
6. The language of the article needs improvement.
Reviewer 3 Report
The references are old, especially in the Discussion section.
Figures are presented in poor quality. Significances are not marked on Figure 5. I suggest making figures in color.
Table 1 is not necessary, because it repeats the text.
In summary, presented work is of interest and may be published after minor revision.
Reviewer 4 Report
Research work presented in the manuscript is good but needs some corrections

Round 2
Reviewer 1 Report
The author addressed the suggestions appropriately
Author Response
We appreciate your valuable time in reviewing our paper and providing valuable comments. We hope that the findings will provide new insights in the field of bioactive peptides.